# Phosphate, Microbiota and CKD

**DOI:** 10.3390/nu13041273

**Published:** 2021-04-13

**Authors:** Chiara Favero, Sol Carriazo, Leticia Cuarental, Raul Fernandez-Prado, Elena Gomá-Garcés, Maria Vanessa Perez-Gomez, Alberto Ortiz, Beatriz Fernandez-Fernandez, Maria Dolores Sanchez-Niño

**Affiliations:** 1Department of Nephrology and Hypertension, IIS-Fundacion Jimenez Diaz, Universidad Autonoma de Madrid, Av Reyes Católicos 2, 28040 Madrid, Spain; chiara.favero.95@gmail.com (C.F.); sol.carriazo@quironsalud.es (S.C.); leticuarental@gmail.com (L.C.); raul.fernandezp@quironsalud.es (R.F.-P.); elena.goma@quironsalud.es (E.G.-G.); MVanessa@fjd.es (M.V.P.-G.); 2Red de Investigacion Renal (REDINREN), Av Reyes Católicos 2, 28040 Madrid, Spain; 3School of Medicine, Department of Pharmacology and Therapeutics, Universidad Autonoma de Madrid, 28049 Madrid, Spain

**Keywords:** chronic kidney disease, microbiota, phosphate, uremic toxins, phosphate binder, short chain fatty acid, PTH

## Abstract

Phosphate is a key uremic toxin associated with adverse outcomes. As chronic kidney disease (CKD) progresses, the kidney capacity to excrete excess dietary phosphate decreases, triggering compensatory endocrine responses that drive CKD-mineral and bone disorder (CKD-MBD). Eventually, hyperphosphatemia develops, and low phosphate diet and phosphate binders are prescribed. Recent data have identified a potential role of the gut microbiota in mineral bone disorders. Thus, parathyroid hormone (PTH) only caused bone loss in mice whose microbiota was enriched in the Th17 cell-inducing taxa segmented filamentous bacteria. Furthermore, the microbiota was required for PTH to stimulate bone formation and increase bone mass, and this was dependent on bacterial production of the short-chain fatty acid butyrate. We review current knowledge on the relationship between phosphate, microbiota and CKD-MBD. Topics include microbial bioactive compounds of special interest in CKD, the impact of dietary phosphate and phosphate binders on the gut microbiota, the modulation of CKD-MBD by the microbiota and the potential therapeutic use of microbiota to treat CKD-MBD through the clinical translation of concepts from other fields of science such as the optimization of phosphorus utilization and the use of phosphate-accumulating organisms.

## 1. Chronic Kidney Disease (CKD) Concept and Global Impact

Chronic kidney disease (CKD) is currently defined as persistent (at least three months) evidence of decreased kidney function or kidney injury [1]. Decreased kidney function is diagnosed by an estimated glomerular filtration rate (eGFR) below 60 mL/min/1.73 m^2^, while the main, but not only criterion for kidney injury is pathological albuminuria. CKD is associated with an increased risk for premature all-cause and cardiovascular death and of progression to kidney failure requiring kidney replacement therapy [1,2]. CKD is one of the fastest growing causes of death, projected to become the fifth global cause of death by 2040 and the second before the end of the century in some countries with long life expectancy [3,4]. It is also the most common risk factor for lethal coronavirus disease 2019 (COVID-19) [5]. Factors involved in the adverse health outcomes associated to CKD range from decreased kidney production of anti-aging factors, such as Klotho, to accumulation of uremic toxins that are not properly excreted by the kidneys [6,7]. The diet is a key source of uremic toxins or of molecules that are processed by the gut microbiota to generate precursors of uremic toxins [8]. Excess dietary phosphate behaves as a uremic toxin that needs kidney excretion. As kidney function is lost, the physiological adaptation to decreased phosphate filtration is one the drivers of CKD-mineral and bone disorder (CKD-MBD) [9]. We now provide a comprehensive holistic and up-to-date review of the bidirectional relationship of phosphate and phosphate binders to the gut microbiota in the context of CKD and explore the potential contribution of this interaction to the gut microbiota-CKD crosstalk [10]. Specifically, we discuss the modulation of CKD-MBD by uremic toxins of bacterial origin, the impact of dietary phosphate and phosphate binders on the gut microbiota, the interaction between vitamin D and parathyroid hormone (PTH) with the microbiota and the potential therapeutic use of microbiota in CKD-MBD through the concepts of optimization of phosphorus utilization and phosphate accumulating organisms. We believe this is the first holistic approach to the topic as a 27 March 2021 PubMed search of the terms phosphate, microbiota and kidney did not disclose any similar review.

## 2. CKD-Mineral and Bone Disorder (CKD-MBD)

CKD-mineral and bone disorder (CKD-MBD) is a systemic disorder triggered by CKD and associated with bone disease, cardiovascular calcification and increased morbidity and mortality [11,12]. It is diagnosed in the presence of biochemical abnormalities of serum calcium, phosphate or PTH, bone abnormalities or cardiovascular or other soft-tissue calcification as a consequence of CKD [13]. Adaptation to a lower phosphate filtration as GFR decreases together with albuminuria- or inflammation-driven decrease in tubular cell Klotho expression are key drivers of CKD-MBD [14,15]. Klotho is a transmembrane or secreted protein expressed mainly by kidney tubules that is a coreceptor for the phosphaturic hormone fibroblast growth factor 23 (FGF23) [16]. The early decrease in kidney Klotho expression and the need to excrete phosphate lead to a progressive increase in serum FGF23 levels as GFR decreases. In proximal tubules, FGF23 decreases phosphate reabsorption, thus promoting phosphaturia, and decreases calcitriol synthesis. Decreased calcitriol availability will limit phosphate and calcium absorption in the gut. Additionally, Klotho has FGF23-independent nephroprotective and anti-aging effects and excess FGF23 in the presence of low Klotho levels has adverse cardiovascular off-target effects. The lower calcitriol concentration and the trend towards hypocalcemia due to decreased gut calcium absorption promote PTH secretion and hyperparathyroidism. These adaptive responses cause bone injury but prevent hyperphosphatemia (normal serum phosphate range 2.5–4.5 mg/dL) until GFR is very low, at which point dietary phosphate restriction and phosphate binders must be prescribed to limit phosphate absorption in the gut and, thus, limit the adverse impact of hyperphosphatemia. Indeed, hyperphosphatemia increases the severity of secondary hyperparathyroidism, leading to bone disease, vascular calcification, and increased incidence of cardiovascular events and mortality [17,18]. In this regard, hyperphosphatemia and the parallel increase of FGF23 are respectively involved in the onset of vascular calcification as well as left ventricular hypertrophy [19].

## 3. Phosphate and CKD

Restricting dietary phosphate implies changing the diet while phosphate binder prescription implies ingesting molecules that bind phosphate and potentially other nutrients, while releasing other components to the gut lumen. While both maneuvers may theoretically modulate the gut microbiota, this has received scarce attention until recently [20].

### 3.1. Diet

In early CKD (GFR > 60 mL/min/1.73 m^2^), there are no randomized clinical trials that support dietary phosphate restriction. However, Western diets contain 2- to 4-fold more phosphate than the daily dietary reference intake for adults of 580 mg or the recommended dietary allowance of 700 mg [21,22,23]. As pathological albuminuria, which usually defines early CKD stages, is already associated with increased serum phosphate, likely as a consequence of suppressed Klotho levels, it would make sense to at least avoid excess dietary phosphate from very early in the course of CKD [14,15,24].

In patients with GFR < 60 mL/min/1.73 m^2^, Kidney Disease: Improving Global Outcomes (KDIGO) guidelines recommend reducing elevated serum phosphate levels to the normal range by using phosphate binders and by limiting dietary phosphate intake [13]. Dietary phosphate is largely derived from high protein foods or food additives. Phosphate from food additives is more readily (around 100%) absorbed, as it is in inorganic form. By contrast, only 50% of organic phosphate in vegetables and around 70–80% of organic phosphate in animal protein-rich products is absorbed. It is suggested that patients with CKD avoid excess dietary protein, mix both animal and vegetable protein, and avoid processed foods rich in phosphate-containing additives [17].

### 3.2. Phosphate Binders

Phosphate binders are the mainstay of pharmacological therapy for hyperphosphatemia in CKD patients [13,25]. They are usually prescribed at advanced CKD stages when hyperphosphatemia is evident, but there is an ongoing reexamination of the potential benefits of ‘preventive’ treatment of early phosphate overload in CKD patients [26,27]. Well designed, larger, long-term clinical trials are needed to respond this critical question. Binders prevent phosphate absorption within the gut by binding an active cation to phosphate, in exchange for another anion (e.g., carbonate, acetate, oxyhydroxide, citrate) to yield a non-absorbable compound excreted in feces [25]. They are classified based on the active cation composition into calcium-containing and calcium-free binders (Table 1). Calcium-free binders may contain magnesium, metals or polymers. A key issue is patient compliance, given the size and number of pills (although there are alternative presentations for some binders) and the common gastrointestinal adverse effects. More than 75% of hemodialysis patients adhered incompletely to phosphate binder prescription in a short follow-up study [28].

Aluminum-containing phosphate binders are effective and well tolerated, but their use is discouraged since the 1980s as they facilitate aluminum intoxication [13]. Calcium-based phosphate binders became popular in the 1980s and 1990s, but they may induce positive calcium balance, aggravate vascular calcification and increase mortality compared to calcium-free binders [25,29]. The fact that they also provide a source of alkali (carbonate, acetate) may compound the problem as a higher pH favors vascular calcification [30]. However, they are still widely used because of lower cost than more modern binders. High dose calcium (in early clinical trials up to 17 g/day of calcium carbonate were prescribed) may also cause precipitation of bile and fatty acids in the form of soap and decrease the absorption of fat-soluble vitamins such as vitamin D and the microbiota metabolite vitamin K, which inhibits vascular calcification [31,32]. However, current guidelines recommend against high-dose calcium-based binders [13,33]. Magnesium carbonate results in a lower calcium load and improved gastrointestinal tolerability [25]. Moreover, magnesium interferes with hydroxyapatite crystal formation and in rats, magnesium carbonate reduced both serum phosphate and aortic calcification [34].

Sevelamer, the first non-metal-containing phosphate binder, launched in 2001, is a large cross-linked cationic polymer and non-absorbable anion exchange binder available as sevelamer carbonate or hydrochloride [25]. In addition to phosphate, sevelamer may bind endotoxins, gut microbiota-derived metabolites, advanced glycation end products and bile salts, consequently decreasing serum low-density lipoprotein (LDL) cholesterol and inflammatory markers in dialysis patients [25,35,36,37,38]. Of these, the most prominent effect is the reduction in LDL-cholesterol, to the point that this is used by clinicians to assess compliance. There are multiple studies addressing the potential consequences of sevelamer binding molecules beyond phosphate. Sevelamer may interfere with absorption of fat-soluble vitamins (D, K) [31,36] and has been associated with evidence of vitamin K deficiency, such as menaquinone MK4 deficiency and increased non-phosphorylated uncarboxylated matrix-Gla protein (dp-ucMGP) [39,40]. These defects were recently described in a cross-sectional study in association with a wider disturbance in uremic toxins of microbiota origin, including increased serum indoxyl sulfate (IS) and phenylacetylglutamine (PAG) in end stage kidney disease (ESKD), mostly dialysis, patients. The mechanisms remain unclear and changes in microbiota composition were hypothesized [39]. In clinical trials, sevelamer carbonate for 12 weeks did not change IS, p-cresyl sulfate (pCS), or indole acetic acid (IAA) levels in non-dialysis CKD patients and in vitro binding to microbiota-derived precursors p-cresol and indole was not observed while it bound IAA at high pH in certain experimental conditions [41]. Sevelamer HCl was previously associated with increased p-cresol (likely reflecting pCS or p-cresyl-glucuronide, see below) but not to changes in IAA or IS in a prospective hemodialysis study [42]. However, it was also associated with reduced p-cresol in non-dialysis patients [43,44,45]. It is unclear whether the molecular form of sevelamer or the different baseline uremic toxin levels in different CKD populations may have accounted for the different results reported. Bixalomer, only currently available in Japan, is another non-absorbable polymer.

Lanthanum has gastrointestinal adverse effects and concerns have been raised regarding bone accumulation in dialysis patients [13,25,46]. It may also interfere with vitamin K absorption [31].

Iron-based phosphate binders are the most recent approach to hyperphosphatemia [47]. Some of them such as ferric citrate, are a source of absorbable iron that improves iron parameters in CKD patients [25]. By contrast, sucroferric oxyhydroxide is a polynuclear chewable iron-based phosphate binder that releases minimal amounts of iron in the gut, resulting in less gastrointestinal side effects and potentially less iron absorption and less impact on the gut microbiota [25,48]. In this regard, iron-based binders may change the gastrointestinal microbiota as gut bacteria may use iron or the organic ligand [36].

## 4. The Microbiota and Biological Impact on CKD

The gut microbiota processes dietary components and secretes bioactive molecules that are absorbed into the circulation. These include vitamins, short-chain fatty acids (SCFAs) and precursors of uremic toxins, all of which may modify CKD progression or CKD manifestations (Figure 1).

### 4.1. Microbiota: Concept and Broad Classification

The microbiota is a complex population of microorganisms (mostly bacteria, but also virus, archaea, and eukaryotes) that contribute to host homeostasis and disease. The bulk (70%) of the microbiota inhabits the gastrointestinal tract [49]. Around 10^14^ microorganisms live in the gut, 10-fold more than the number of human cells, and, collectively, express 250–800 times more genes than the human genome [50]. The gut microbiota contributes to host health by promoting gut integrity [51], regulating host immunity [52], protecting against pathogens [53] and providing nutrients and bioactive molecules such as vitamins and SCFAs. However, the composition of the gut microbiota may be altered by drugs, diet, stress and disease. An altered microbiota (the so-called “dysbiosis”) may contribute to disease by failing to contribute to a healthy host–microbiota interaction or by actively promoting the disease state through the production of microbial metabolites. Dysbiosis is frequently characterized by decreased microbial diversity and an increase in specific taxa [54,55]. The healthy gut microbiota is composed of up to 1000 different microorganisms. Despite interindividual variability and intraindividual variability over time, it seems that a functional core microbiome is common to human hosts of different gender, age and geographic location. The most abundant bacterial phyla are *Bacteroides* and *Firmicutes* that constitute approximately 90% of the colonic/fecal microbiota., followed by *Actinobacteria, Verrucomicrobia* and a lower presence of *Proteobacteria* [56]. *Firmicutes* and *Bacteroides* are carbohydrate fermenters and produce a pool of fatty acids that are used as an energy source by the host. Besides, *Bacteroides* express polysaccharide A, which can induce regulatory T cell growth and cytokine expression. [57].

### 4.2. Bioactive Molecules Released by the Microbiota

The microbiota releases bioactive metabolites that modulate health and disease [58]. The specific metabolites and quantity of metabolites released depends on the overall status of the microbial community: composition, species diversity and diet composition [59]. Bioactive molecules released by microbiota include SCFAs (e.g., butyrate, propionate, acetate and crotonate), gases (hydrogen, methane, carbon dioxide and hydrogen sulphide), polyamines, polyphenols, and vitamins derived from the microbial fermentation of undigested nutrients [60]. Bile acids are metabolized through deconjugation and dihydroxylation reactions by microbial enzymes such as BSH (bile salt hydrolase) and BAI (bile acid-inducible) that are fundamental for bile acid homeostasis [53].

A growing body of literature has focused on the effects of microbial SCFAs on the host [58]. Butyrate represents the primary energy source for colonocytes. Additionally, butyrate modulates the epigenetic regulation of gene expression by inhibiting histone deacetylase (HDAC) with resulting anti-cancer and anti-inflammatory properties [58,61,62]. Butyrate also binds to FFAR2/GPR43, FFAR3/GPR41 and GPR109A. Propionate is absorbed and rapidly metabolized by the liver whereas acetate is the most abundant SCFA in the peripheral circulation [63], crosses the blood–brain barrier and controls appetite [64]. Acetate is an intermediary metabolite integrated in the Krebs cycle as acetyl-CoA that may also behave as a ligand for GPR43 and GP341 expressed in gut, adipose tissue, liver and pancreas, as an epigenetic regulator and as modulator of AMP-activated protein kinase (AMPK) activity [65,66]. Through these actions it may improve glucose and lipid metabolism and increase fatty acids synthesis, among others. Finally, crotonate also modulates epigenetic regulation of gene expression through histone crotonylation and has anti-inflammatory and kidney protective properties [67,68,69].

SCFA have been related to the pathogenesis of kidney injury and phosphate balance. Thus, inhibition of HDACs reversed the negative impact of albuminuria and inflammatory cytokines on the expression of Klotho, the key co-receptor for the phosphaturic hormone FGF-23 that additionally has kidney protective effects [15,70]. Specifically, butyrate preserved Klotho expression by this mechanism [71]. Histone crotonylation preserved the expression of the master regulator of mitochondrial biogenesis peroxisome proliferator-activated receptor gamma coactivator-1α (PGC-1α), which contributes to preserve proximal tubule cell function, decrease kidney inflammation and preserve kidney function [69,72,73]. Additionally, either dietary fiber-induced production of SCFA or treatment with SCFA protected from experimental diabetic nephropathy in a manner dependent on GPR43 and GPR109A expression [74].

SCFA derive from dietary fiber fermented by the gut microbiota under anaerobic conditions [75]. In this regard, the type and quantity of dietary fermentable fibers influences the composition of the microbiota and SCFA production. Specifically, dietary fiber modulates the *Firmicutes* to *Bacteroides* ratio [76]. Acetate is synthesized either via acetyl-CoA or the Wood–Ljungdahl pathway [77]. Propionate derives from the succinate, acrylate or propanediol pathways [78,79]. Butyrate is produced from two molecules of acetyl-CoA that are converted to acetoacetyl-CoA. In turn, this product is converted to butyryl CoA, via the intermediates L(+)-β-hydroxybutyryl-CoA and crotonyl-CoA. Subsequently, butyryl CoA is transformed into butyrate by either butyrate kinase (e.g., *Coprococcus comes*, *Coprococcus eutactus*) or butyryl-CoA:acetyl-CoA transferase (e.g., *Faecalibacterium prausnitzii*, *Eubacterium rectale*, *Eubacterium hallii*) [80,81]. The biological impact of SCFA was recently emphasized by a study in pregnant mice, in which dietary fiber modulation of SCFA production by microbiota resulted in long-term metabolic consequences in the newborn through activation of SCFA receptors [82]. Specifically, propionate activation of GPR43 and GPR41 signaling was a key mediator.

### 4.3. Uremic Toxin Precursors Released by the Microbiota

Amino acids may be metabolized by the gut microbiota into precursors of uremic toxins which are then converted to uremic toxins in the liver [83,84]. Gut-derived uremic toxins include trimethylamine N-oxide (TMAO), pCS, IS, IAA, hippuric acid, p-cresyl glucuronide, phenyl acetyl glutamine and phenyl sulfate [85]. The biological activity of TMAO, pCS and IS has been characterized in most detail and found to promote vascular and kidney injury and to engage proinflammatory and profibrotic pathways [86,87].

**TMAO** is generated in the liver from trimethylamine (TMA), which in turn is generated from choline and carnitine by TMA lyase, a microbiota enzyme. Plasma TMAO levels are associated with cardiovascular events [88]. TMAO promoted atherogenesis and kidney tubulointerstitial fibrosis by recruiting Runx2 and bone morphogenetic protein 2 (BMP2) in vascular smooth muscle cells, driving osteogenic differentiation, and activation of the NLRP3 inflammasome, nuclear factor kappa-B (NFκB) and transforming growth factor-β1/small mother against decapentaplegic (TGF-β1/SMAD) signaling [89,90,91]. Although Runx2 has been best characterized in the vascular calcification context, it also promoted kidney injury [92]. However, whether TMAO also engages kidney Runx2 has not been addressed. Organic cation transporter 2 (OCT2) mediated reabsorption of TMAO by kidney proximal tubular cells, thus potentially exposing these cells to the toxin [93,94]. TMAO levels may be reduced by lowering excess dietary choline and carnitine, which are components of diverse dietary supplements. Antibiotics may also decrease TMAO levels but at the expense of impairing microbiota homeostasis [91]. More recently, iodomethylcholine, a TMA-lyase inhibitor, reduced TMA and TMAO generation by selectively inhibiting TMA generation by the microbiota, attenuating kidney tubulointerstitial fibrosis and kidney function loss [95].

**pCS and IS** are derived from the microbiota metabolism of tyrosine and tryptophan to p-cresol and indole, respectively. Serum pCS and IS were increased in CKD patients and were associated major cardiovascular events [96,97,98]. They circulate as protein-bound molecules, mainly bound to albumin. Protein-binding decreased glomerular filtration and, thus, active tubular secretion was required for excretion [99]. Protein-binding also impaired clearance by dialysis. While p-cresol toxicity has been widely studied [100,101,102], p-cresol is rapidly metabolized to pCS or p-cresyl glucuronide and initial reports of high free p-cresol levels in CKD may have resulted from methodological issues [103]. In general, p-cresyl glucuronide was less toxic than pCS [104,105]. pCS and IS induced inflammatory responses in cultured proximal tubular cells [105,106,107], epithelial to mesenchymal transition and profibrotic TGF-β1 and Snail upregulation [108,109,110,111,112,113,114,115,116], kidney epidermial growth factor receptor (EGFR) activation and tubulointerstitial expression of matrix metalloproteinases 2 and 9 [117], oxidative stress through reduction of glutathione levels [118] and mitochondrial dysfunction through induction of mitochondrial fission proteins, reducing biogenesis and decreasing mitochondrial mass due to excess autophagy [119]. Additionally, pCS caused endothelial barrier dysfunction due to vascular endothelial (VE)-cadherin phosphorylation by Src and decreased intercellular junctions [120,121,122]. Decreasing IS and pCS may improve outcomes in CKD patients, although this is difficult to demonstrate as therapeutic maneuvers that decrease these toxins may also have other targets. AST-120 binds uremic toxin precursors in the gut and is used in Japan to delay CKD progression [123]. However, clinical trials outside Japan failed to confirm efficacy. Compliance may have been an issue, given the high number of daily pills required (a total of 30 capsules daily) [124]. Additionally, the sulphotranspherase inhibitors resveratrol and quercetin prevented the liver synthesis of IS, suppressed IS accumulation and improved kidney function in experimental AKI [125]. However, these drugs have additional targets and since kidney function improved, it is unclear whether IS levels decreased due to decreased synthesis or increased excretion.

**Microbiota-derived uremic toxins and CKD-MBD.** TMAO, IS and pCS may modulate CKD-MBD components by promoting vascular calcification and/or interfering with bone health by suppressing bone formation and bone resorption [91,126,127,128,129]

**Potential interventions.** In CKD patients, the gut generation of uremic toxin precursors has been reported to be stable, but the microbial species that generate SCFA were reported to be decreased [86,130,131]. A metabolome study identified lower serum 3-indolepropionic acid (IPA) levels in 10 patients with rapid kidney function decline than in 10 controls. In a validation cohort, serum IPA levels in 140 CKD patients were lower than in 144 healthy individuals 34.7 vs. 49.8 ng/mL, *p* < 0.01 [132]. This was contrary to the high levels of other microbiota-derived metabolites, such as IS and pCS, as previously described for CKD. Interestingly, IPA is also a tryptophan metabolite produced by *Clostridium sporogenes* that has been considered as a healthy microbiota marker, as it was increased in individuals with high dietary fiber intake, has potent oxygen radical scavenging properties, activates the pregnane X receptor (PXR) (thus decreasing intestinal permeability) and has been associated with neuroprotection and with a lower risk for type 2 diabetes [133]. These findings should be confirmed in larger cohorts from different continents but illustrate the potential for microbiota-targeted interventions. Preserving bacteria responsible for SCFA generation and decreasing those species that promote uremic toxin synthesis would be a key aim of treatment [134,135]. For example, *L. salivarius* BP121 protected from cisplatin-induced AKI and suppressed IS and pCS production [136]. Prebiotics, such as xylooligosaccharide may also promote SCFA synthesis and decrease uremic toxins [137,138,139]. However, none of these therapeutic approaches is routinely used in the clinic, pending large scale clinical trials. Prescription of low potassium diets to manage CKD-associated hyperkalemia is not expected to be helpful for optimal SCFA synthesis, as these diets are usually also low in fiber [140,141].

## 5. Impact of Dietary Phosphate on Gut Microbiota

The cellular and molecular pathways involved in the association between increased dietary phosphate intake and increased mortality even in healthy Americans are unclear and little is known about the impact of dietary phosphate intake on the gut microbiota [142]. Since the excess dietary phosphate of Western diets is to a great extent due to the intake of food and beverages rich in phosphate-containing additives, excess dietary phosphate is associated with potential confounders regarding lifestyles that may also impact on mortality [8]. Hence, it is fundamental to understand every potential target of dietary phosphate in the body, including how dietary phosphate interacts with the gut environment and microbiota.

Phosphate is a prebiotic that influences the growth of microbes [143]. Interventional studies have evaluated the consequences of calcium and phosphate supplementation on the gut microbiota. Supplementation of both led to amorphous calcium phosphate (ACP) precipitates in the intestinal lumen under certain conditions. ACP precipitated bile acids and fatty acids through hydrophobic aggregation [144]. A decreased bile acid availability can modulate the microbiota and fatty acid precipitation may modulate the biological impact of microbial products. Calcium-phosphate nanoparticles in the gut also modulate immune tolerance against the microbiota [145]. A recent study analyzed fecal samples of healthy individuals on different doses of phosphate and calcium supplements: 1000 mg phosphate, 1000 mg phosphate/500 mg calcium and 1000 mg phosphate/1000 mg calcium. In men, the latter dose shifted the microbial community compared to the other groups and increased acetate concentration, likely a consequence of the higher presence of Clostridium, which release acetate from carbohydrate fermentation [144]. The combination of calcium carbonate and phosphate supplementation increased fecal excretion of SCFA, including acetate and propionate, but not of butyrate. These changes were not observed in women but the sample size was small. Interestingly, phosphate intake of this magnitude may be found in the general population and in CKD patients (some artificial beverages contain up to 630 mg per serving) [146] and a single dose of calcium carbonate used as a phosphate binder already contains 400 mg calcium while the recommended dietary allowance of calcium is 1000 to 1200 mg in adults [21]. However, studies addressing the impact of different relative amounts of calcium and phosphate on the microbiota in persons with CKD are lacking.

Another study evaluated the microbial composition of fecal samples of persons with prolonged stays in intensive care units (ICU). During stays in the ICU, microbial diversity can be lost, and resistance genes can be selected. Most ultra-low-membership communities had low virulence when grouped together, however they showed a harmful behavior when isolated. In ICU patients, the gut microbiota could be formed by ultra-low-diversity communities of multidrug-resistant pathogens and the shift from low to high virulence might be caused by opioids released during critical illness. Phosphate-polyethylene glycol [Pi-PEG] behaved as an antivirulence agent by increasing local phosphate availability and prevented opioid-induced virulence. Thus, local gut phosphate can drive the behavior of bacterial communities [147].

There is more literature on farm animals, in which the question addressed is how to best increase phosphate utilization minimizing mineral phosphate supplements. Phosphate supplementation in pigs influenced the composition and activity of the microbiota in interaction with dietary calcium, modifying SCFA production [148,149,150]. Individual phosphate-containing molecules or phosphate may also affect properties of pathogenic micro-organisms, such as metabolism or virulence [150]. A high calcium/phosphate content negatively affected the intestinal abundance of certain fermentation end-products (e.g., reduced propionate in the large intestine) in weaned pigs [151]. In growing pigs, ileal bacterial populations and fermentation patterns changed with the intestinal availability of calcium and phosphate. Higher calcium availability in the gut reduced the numbers of some Gram-positive bacteria, whereas high phosphate availability increased the growth of strictly anaerobic bacteria [152]. Also, in broilers, feeding different amounts of phosphate, calcium and phytase (used to increase phosphate availability from plant sources) led to a shift in gut microbiota [148].

In summary, although not conclusive, these studies provided first hints suggesting that dietary phosphate modulates the gut microbiota composition and behavior, potentially having systemic effects so far understudied. The impact of calcium interactions has further significance to the CKD situation in patients treated with calcium-containing phosphate binders.

## 6. Impact of Phosphate Binders on Gut Microbiota

Besides reducing dietary phosphate absorption, phosphate binders may alter the gut microbiota [36]. The effects of phosphate binders on gut microbiota may depend on the specific compound used. Several mechanisms have been described including decreased phosphate availability and resultant decrease of phosphate-dependent bacteria, binding of non-phosphate molecules, release of iron or formation of phosphate complexes (e.g., ACP) that may in turn precipitate other bioactive intestinal molecules that modulate bacterial biology [144,153,154].

As discussed in the prior section, in healthy individuals, calcium carbonate supplementation modified the gut microbiota and increased fecal excretion of SCFA [144]. However, these changes were only observed in men and not in women, and CKD patients were not studied. In hemodialysis patients, calcium carbonate was associated with a significant increase of the gut microbial dysbiosis index and a reduction of microbial species diversity when compared to ferric citrate. Ferric citrate was associated with an increased abundance of *Bacteroides* and decreased abundance of *Firmicutes*. Members of the order Lactobacillales were enriched in patients treated with calcium carbonate, whereas taxa of the genera *Ruminococcaceae UCG-004*, *Flavonifractor*, and *Cronobacter* were enriched in patients treated with ferric citrate [155].

There is inconsistent evidence regarding sevelamer and microbiota, potentially dependent on species, baseline clinical conditions, and length of follow-up, among other confounders. In mice with non-alcoholic steatohepatitis, sevelamer reversed and prevented progression of liver injury in association with changes in the microbiota population and bile acid composition, including reversing microbiota complexity in the cecum by increasing *Lactobacillus* and decreasing *Desulfovibrio* [156]. However, to what extent the microbiota changes were primary or secondary to improved liver injury is unclear from the experimental design. In this regard, no changes in fecal microbiota were observed in a pilot double-blinded controlled trial of persons with type 2 diabetes randomized to sevelamer or placebo [157].

Lanthanum carbonate administration for 12 weeks to hemodialysis patients was associated with reduced diversity of the gut microbiota characterized by decreased *Parvimonas*, *Gemella*, *Centipeda*, *Chryseobacterium*, *Pelomonas*, *Curvibacter* and unclassified Rhodocyclaceae and changes in their network complexity [153].

Iron contained in phosphate binders could shift the gut microbiota as oral iron supplements do. This specially applies to ferric citrate which is associated with increased iron availability, as discussed above for the calcium carbonate/ferric citrate comparison [47]. In subtotally nephrectomized rats, 4% ferric citrate increased fecal alfa diversity, while the relative abundance of Firmicutes decreased and the relative abundance of the *Akkermansia* genus and the Clostridiaceae and Enterobacteriaceae families increased, compared with untreated CKD rats [158]. Ferric citrate also increased levels of Tryptophanase-possessing bacteria, which are linked to indole and p-cresol production. However, plasma levels of these molecules were not increased. The altered gut microbiota was associated to improved kidney function, adding a further source of confusion when interpreting microbiota changes [158].

Overall, the impact of phosphate binders on gut microbiota is starting to be recognized but is still incompletely understood. Detailed clinical studies are needed to characterize their impact on gut microbiota, host exposure to SCFAs and precursors of uremic toxins, and potential relation of this interaction with outcomes. Modulation of gut microbiota by phosphate binders may contribute to the observed results in clinical studies assessing the impact of phosphate binders on circulating levels of uremic toxins of gut origin, beyond any direct interaction of binders with these toxins or their precursors.

## 7. Impact of Gut Microbiota on CKD-MBD

There is also evidence that the gut microbiota may modulate CKD-MBD.

**Vitamin D.** Vitamin D promotes calcium and phosphate absorption in the gut, regulates immune functions and influences the gut microbiota [159,160,161]. Vitamin D deficiency in newborn mice changed microbiota composition later in life, and following bacterial infections, dysbiosis was more prominent in patients with vitamin D deficiency [161]. Moreover, vitamin D deficiency decreased gut vitamin D receptor (VDR) expression [162] and this, in turn, generated dysbiosis [163]. However, there is less information on how the microbiota modifies vitamin D availability or activity. A microbial metabolite generated from bile acids, lithocholic acid (LCA), activated the VDR [164]. LCA may compete with vitamin D for VDR binding and, for example, trigger calcitriol degradation or activate VDR to modulate Treg populations [165]. Additionally, SCFA may increase VDR expression and hydroxylases from some bacteria may activate vitamin D (reviewed in [166]).

**Parathyroid hormone (PTH).** PTH modulates bone remodeling, promoting both bone formation and bone resorption depending on whether bone cells are exposed to PTH continuously or intermittently [167]. The gut microbiota may modulate PTH impact on bone health. Butyrate produced by the gut microbiota contributes to gut-bone communication in young female mice. Butyrate activation of GPR43 signaling in dendritic cells and GPR43-independent signaling in T cells increased Treg differentiation in bone marrow, contributing to PTH anabolic activity in bone. Moreover, nutritional supplementation with the probiotic *Lactobacillus rhamnosus GG* (LGG) increased butyrate production and stimulated bone formation [168].

Besides its contribution to CKD-MBD, hyperparathyroidism is one of the causes of osteoporosis. In murine hyperparathyroidism, PTH caused bone loss in mice colonized with Th17 cell-inducing taxa segmented filamentous bacteria (SFB). Indeed, bone loss may be induced by TNF generated in the small intestine in response to bacterial products, such as flagellins or LPS. In mice with a depleted microbiota, PTH administration and low calcium diet did not induce bone loss, but SFB was sufficient for PTH to exert the catabolic activity. In this regard, trafficking of TNF + T cells and Th17 cells from the gut to the bone marrow was required for PTH-induced bone loss [169]. These results provide insights into novel therapeutic approaches to bone loss by either supplementing with probiotics [168] or targeting the microbiota and blocking T cell migration [169]. However, it is unknown whether these mechanisms are active in or modified by CKD.

Phosphate. The gut microbiota may also modify the phosphate balance. In a placebo-controlled clinical trial, oral administration of *B. longum* decreased serum phosphate in hemodialysis patients. It was proposed that probiotics increased calcium ionization, favoring calcium binding to phosphate and, thus, reducing phosphate absorption [170].

Additionally, research in farm animals is contributing to increase our understanding of the interactions between different sources and amounts of phosphate and the gut microbiota. The problem to be solved in farm animals is how to increase phosphate utilization, that is, how to increase the phosphate-accretion/phosphate-intake ratio to minimize the need to supplement dietary phosphate. However, insights obtained in studies aimed at understanding how to maximize phosphate absorption may also provide insight into what maneuvers result in lower phosphate absorption. Phosphorus utilization is a heritable tract in farm animals that may be influenced by the gut microbiota. In Japanese quail, seven of 55 genera of microbiota were more abundant in high phosphorus utilization animals and *Bacteroides* played an important role in phosphorus utilization [148]. In this regard, inositol phosphates (InsPx) are the major source of phosphate in plant seeds and need phytase activity to release phosphate. This may be provided by microbes as in non-ruminant, monogastric animals, phytase activity is very low. Molecular adaptations to low-phosphate diets might contribute to the cleavage of phosphate from InsPx and enhance phosphate absorption in the gastrointestinal tract [148]. Understanding these molecular adaptations and how to combat them may contribute to design novel approaches to further decrease phosphate absorption in CKD patients.

## 8. New Tools: Phosphate-Accumulating Organisms

Phosphate-accumulating organisms (PAOs) are specific microbes that accumulates phosphate in the form of polyphosphate. PAOs are now under intense scrutiny as a potential method to extract phosphate from wastewater in a manner that can allows phosphate recycling [171]. These studies imply the identification and isolation of PAOs and methods to assess polyphosphate contents [171,172,173]. Similar techniques may eventually be used to characterize gut microbiota PAOs that prevent phosphate absorption in the gut in CKD patients, facilitating phosphate excretion in feces. Thus, similar to PAOs removal of phosphate from wastewater, identification of PAOs in the normal gut microbiota, and fostering their growth may promote phosphate accumulation in PAOs. Phosphate accumulated in PAOs in the form of polyphosphate will not be available for absorption and will be excreted in feces, together with the PAOs (Figure 2). For example, Betaproteobacteria-, Cytophagia-, and Chloroflexi-class bacteria were identified as PAO and these classes are present in human guts [173,174]. *Oxalobacter formigenes*, an oxalate-degrading anaerobic bacterium in the human gut of potential interest for the treatment of oxalate urolithiasis and hyperoxaluria, belongs to the Betaproteobacteria class [175]. Interestingly, an interaction between SCFA and PAOs has been described [176]. In this regard, butyrate and propionate support PAOs [176,177]. Still wastewater research is performed at environmental conditions that may not be relevant for the human gut. Thus, adaptation of PAO research findings to gut environment conditions should be specifically pursued.

## 9. Conclusions

The role of the microbiota in the genesis of protein-bound toxin precursors is well established. However, much less is known regarding the interaction between phosphate, another uremic toxin, and the microbiota. Evidence so far indicates that both dietary phosphate and phosphate binders may modulate the microbiota composition and the bioavailability of microbial products such as biologically active SCFAs and vitamin K (Figure 3). Furthermore, the microbiota may modulate dietary phosphate disposal and absorption, as well as CKD-MBD, for example, by modulating PTH actions on bone (Table 2). However, the precise molecular mechanisms for the phosphate-microbiota-host crosstalk and overall clinical impact remain unclear. In this regard, while experimental studies pointed the way, human studies are needed to understand the impact of dietary phosphate on the microbiota and potential clinical consequences and, conversely, the impact of the microbiota on phosphate balance and CKD-MBD manifestations. Unravelling these relationships may help develop novel therapeutic approaches for CKD-MBD that may borrow concepts from other areas of science such as phosphorus utilization PAO research. In this regard, key research questions on the relationship between phosphate and the microbiota and its clinical impact remain to be addressed (Table 3).

## Figures and Tables

**Figure 1 nutrients-13-01273-f001:**
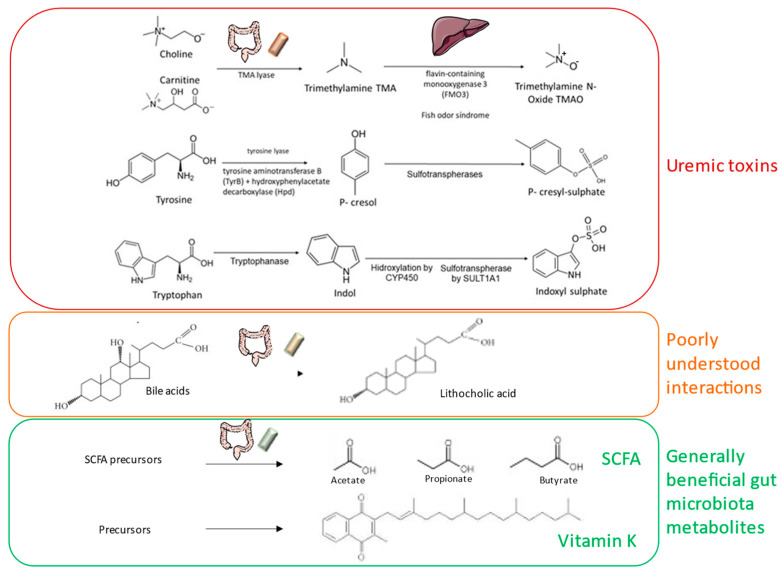
Biologically active molecules produced by the microbiota of interest for chronic kidney disease-mineral and bone disorder (CKD-MBD). SCFA: short-chain fatty acids.

**Figure 2 nutrients-13-01273-f002:**
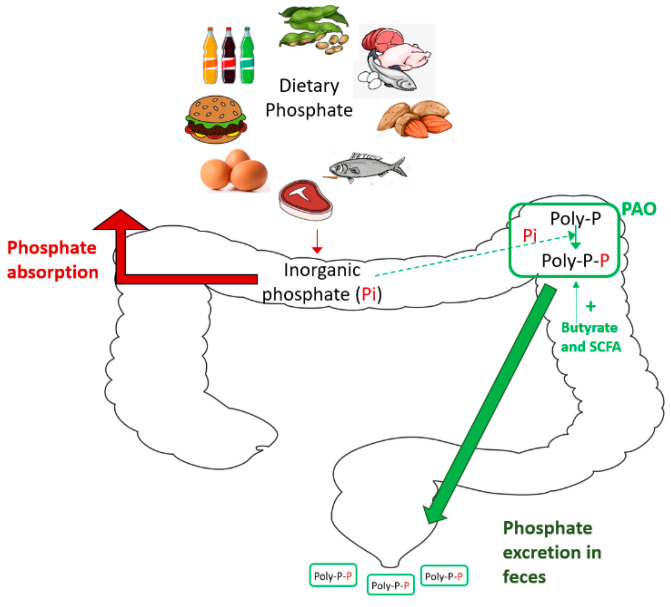
Phosphate-accumulating organisms (PAOs) to prevent positive phosphate balance in CKD.

**Figure 3 nutrients-13-01273-f003:**
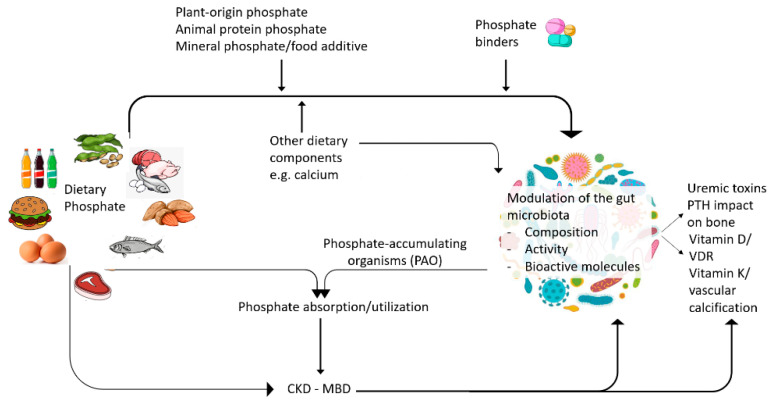
Interactions between dietary phosphate and the microbiota.

**Table 1 nutrients-13-01273-t001:** Key phosphate binders.

Drug	Usual Dose ^1^	Advantages	Disadvantages	Characteristics
**Calcium-based binders**				
Calcium carbonate	500–1250 mg (3–12 tablets)	Effectiveness. Non evidence of influence in gut microbiota.	Hypercalcemia and vascular calcification. Gastrointestinal: constipation, nausea, vomiting.	Election therapy in 1980–1990s. Reduce carboxylation of matrix g-carboxyglutamate protein, a protein that inhibits calcification.
Calcium acetate *	667 mg (6–12 capsules)			
**Magnesium-based binders**				
Magnesium carbonate *	63 mg (2–6 capsules)	Lower calcium overload and vascular calcification. Gastrointestinal tolerability.	Diarrhea. Hypermagnesemia.	Experimental data suggests that magnesium interferes with hydroxyapatite crystal formation.
**Polymeric binders**				
Sevelamer hydrochloride	800–1600 mg every 8 h	Nonproducing calcium overload. Improves endothelial functions. Reduces bile salt absorption	High bill burden. Gastrointestinal tolerability. Interference in absorption of fat-soluble vitamins. High costs.	Exchange of carbonate or HCl for Pi. First non-metal phosphate binder. Large cross-linked cationic polymer.
Sevelamer carbonate	800–1600 mg every 8 h			
Bixalomer	250 mg (6–14 tablets)	Gastrointestinal tolerability. Less water absorption Better fluidity.	Non available.	Amine-functional and non-absorbable polymer. Only in Japan.
**Metal-based binders (non-iron)**				
Aluminum-based	640 mg (5–6 tablets)	Gastrointestinal tolerability.	Aluminum intoxication: encephalopathy, osteomalacia, microcytic anemia and premature death	First available binder Use strongly discouraged by KDIGO guidelines.
Lanthanum carbonate	250–1000 mg (3–6 chewable tablets)	Lower pill burden. Gastrointestinal tolerability.	Accumulation in bone in dialysis patients. Low solubility.	First calcium-free chewable phosphate binder. Detaches carbonate and forms a lanthanum-phosphate complex.
**Metal-based binders (iron)**				
Ferric citrate	210 mg (4–5 tablets)	Lower pill burden. Improves iron parameters.	Gastrointestinal tolerability: diarrhea, nausea, vomiting. Altered gut microbiota	Forms a non-soluble ferric-phosphate complex.
Sucroferric oxyhydroxide	500 mg (2–6 chewable tablets)	Less gastrointestinal effects than ferric citrate. Less alteration of gut microbiota. Lower pill burden.		Polynuclear chewable iron-based phosphate binder.

^1^ Usual dose based on leaflet information and [25]. * Calcium acetate and magnesium carbonate may be combined in a single pill. KDIGO: Kidney Disease: Improving Global Outcomes. Bold identified families of phosphate binders.

**Table 2 nutrients-13-01273-t002:** Key recent findings on phosphate, CKD-MBD and microbiota.

The gut microbiota is a source of beneficial bioactive molecules (e.g., SCFA, IPA, vitamin K)) and of uremic toxin precursors that collectively may impact host health including CKD and CKD-MBD.
The source and amount of dietary phosphate and the dietary calcium:phosphate ratio may modify the gut microbiota composition and properties.
Treatment for CKD-MBD, including phosphate binders, may influence the gut microbiota composition and properties in a binder-specific manner.
The gut microbiota may modulate CKD-MBD through SCFA-mediated modulation of Klotho expression, modulation of vitamin D and PTH activity, thus modulating bone health, serum phosphate and phosphate balance.
Phosphorus utilization research in farm animal research explores how to modulate phosphate uptake from the diet.
Phosphate-accumulating organisms (PAOs) are used in wastewater research to remove phosphate for the microenvironment.
Findings from phosphorus utilization and PAO research may be applied to prevent dietary phosphate absorption in human CKD.

**Table 3 nutrients-13-01273-t003:** Key research questions on phosphate and microbiota.

What is the optimal dietary phosphate intake and optimal form of dietary phosphate from the point of view of a healthy microbiota? In the general population? And in CKD patients?
What is the optimal phosphate binder from the point of vew of a healthy microbiota?
What phosphate binder best promotes the microbiota production of beneficial and bioavailable short chain fatty acids?
What is the optimal phosphate binder to decrease uremic toxins production by the gut microbiota?
What components of the gut microbiota minimize the adverse consequences of CKD-MBD by modulating vitamin D, PTH or other key host activities?
How can we promote and maintain such microbiota? Can dietary interventions, prebiotics, probiotics or symbiotics achieve this?
Are there phosphate-accumulating organisms (PAO) in the gut microbiota that can be used to increase the fecal excretion of dietary phosphate?

## Data Availability

Not applicable.

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
