# Peer review of "Phosphate, Microbiota and CKD"

_nutrients, 2021, doi:10.3390/nu13041273_

Round 1
Reviewer 1 Report
Favero et al. reviewed the relationship between dietary phosphate and phosphate binders, gut microbiota, and CKD-MBD and discussed the potentials of modulating gut microbiota as therapeutic for CKD-MBD. However, the misused terms and inappropriate conclusions in this review seriously affected its readability and significance.
Specific examples are as follows:
L191: Please specify this is the "core phylum" of colonic /fecal microbiota. The microbial composition of other segments/locations is quite different.
L193: Both Firmicutes and Bacteroidetes involve in energy metabolism. Please check this statement carefully.
L196: I would suggest re-write this section by 1) focusing on the "bioactive molecules" that are relevant to phosphate or CKD other than generally talking about SCFAs; 2) adding more information (e.g. potential signal pathways) linking these molecules to CKD. Otherwise, this section is too informative-less.
L198: What's the meaning of "the quality of metabolites"? Please check the statement carefully.
L210: What are the resultant responses of the hots? Please add more information.
L212: I would think one of the most important roles of acetate is acting as intermediates for further bio-synthesis or biological activities. Please check the relevant statements and add more details.
L219: Again, what do you mean by "the quality of dietary fibers"?
L223: I believe there is another pathway for butyrate formation that is mediated by butyrate kinase.
L282: Change "the microbiota" to "the microbial species" or "the microbes".
L283: Please add more information about this statement. What's the concentration of 3-indolepropoinic acid detected in heath individuals? And what's the concentration in CKD patients??? Is this widely detected in clinical trials? If yes, add more details (e.g. sample size...)
L295: Redundant statements. Please simplify the sentences.
L304: Please use terminology correctly: 1) "probiotic property" is used for "probiotics" or "potential probiotics"; 2) "microbiota" includes all the microorganisms in a specific environmental sample. Please use "microbes"/"microorganisms " here.
L327: When describing the findings of a reference, please use the past tense and check for similar issues in this review!
L331: Please be careful with this statement. I would suggest deleting "from commensal to pathogenic lifestyle". Same comments for L334 and L335
Author Response
Favero et al. reviewed the relationship between dietary phosphate and phosphate binders, gut microbiota, and CKD-MBD and discussed the potentials of modulating gut microbiota as therapeutic for CKD-MBD. However, the misused terms and inappropriate conclusions in this review seriously affected its readability and significance.
Specific examples are as follows:
- L191: Please specify this is the "core phylum" of colonic /fecal microbiota. The microbial composition of other segments/locations is quite different.
R: This was changed to “The most abundant bacterial phyla are Bacteroides and Firmicutes that constitute approximately the 90 % of the colonic /fecal microbiota.”
- L193: Both Firmicutes and Bacteroidetes involve in energy metabolism. Please check this statement carefully.
R: This was expanded to “Generally speaking, Firmicutes and Bacteroides are carbohydrate fermenters and may help in the production of a pool of fatty acids that are used as an energy source by the host. Besides, Bacteroides ex-presses polysaccharide A, which can induce regulatory T cell growth and cytokine expression”
- L196: I would suggest re-write this section by 1) focusing on the "bioactive molecules" that are relevant to phosphate or CKD other than generally talking about SCFAs; 2) adding more information (e.g. potential signal pathways) linking these molecules to CKD. Otherwise, this section is too informative-less. And L210: What are the resultant responses of the hots? Please add more information
R: we thank the reviewer for the suggestion. The relevance of SCFA for CKD and phosphate balance is now detailed, together with the pathways identified so far as follows:
“SCFA have been related to the pathogenesis of kidney injury and phosphate balance. Thus, inhibition of HDACs reverses the negative impact of albuminuria and inflammatory cytokines on the expression of Klotho, the key co-receptor for the phosphaturic hormone FGF-23 that additionally has kidney protective effects (29425318, 21719790). Specifically, butyrate preservation of Klotho expression by this mechanism has bene reported (32554208). Histone crotonylation preserves the expression of the master regulator of mitochondrial biogenesis peroxisome proliferator-activated receptor gamma coactivator-1α (PGC-1α), which contributes to preserve proximal tubule cell function, decrease kidney inflammation and preserve kidney function (26535995, 30982966, 27125278). Additionally, either dietary fiber-induced production of SCFA or treatment with SCFA protected from experimental diabetic nephropathy in a manner dependent on GPR43 and GPR109A expression (32358041).”
- L198: What's the meaning of "the quality of metabolites"? Please check the statement carefully.
R: This was changed to “specific metabolites”
- L212: I would think one of the most important roles of acetate is acting as intermediates for further bio-synthesis or biological activities. Please check the relevant statements and add more details.
- We thank the reviewer for this suggestion and the section has been expanded as follows: “Acetate is an intermediary metabolite integrated in the Krebs cycle as acetyl-CoA that may also behave as a ligand for GPR43 and GP341 expressed in gut, adipose tissue, liver and pancreas, as an epigenetic regulator and as modulator of AMP-activated protein kinase (AMPK) activity [64,65]. Through these actions it may improve glucose and lipid metabolism and increase fatty acids synthesis, among others.”
- L219: Again, what do you mean by "the quality of dietary fibers"?
R: This was changed to “type”
- L223: I believe there is another pathway for butyrate formation that is mediated by butyrate kinase.
R: We added this pathway as follows: “Butyrate is produced from two molecules of acetyl-CoA that are converted to acetoacetyl CoA. In turn, this product is converted to butyryl CoA, via the intermediates L(+)-β-hydroxybutyryl CoA and crotonyl CoA. Subsequently, butyryl CoA is transformed into butyrate either by either butyrate kinase or butyryl-CoA:acetyl-CoA transferase”
- L282: Change "the microbiota" to "the microbial species" or "the microbes".
R: Done as suggested
- L283: Please add more information about this statement. What's the concentration of 3-indolepropoinic acid detected in heath individuals? And what's the concentration in CKD patients??? Is this widely detected in clinical trials? If yes, add more details (e.g. sample size...)
R: More information added as suggested as follows: “A metabolome study identified lower serum 3-indolepropionic acid (IPA) levels in 10 patients with rapid kidney function decline than in 10 controls. In a validation cohort, serum IPA levels in 140 CKD patients were lower than in 144 healthy individuals 34.7 vs 49.8 ng/ml, p <0.01 [126]. This was contrary to the high levels of other microbiota de-rived metabolites, such as IS and pCS, as previously described for CKD. Interestingly, IPA is also a tryptophan metabolite produced by Clostridium sporogenes that has been considered as a healthy microbiota marker, as it was increased in individuals with high dietary fiber intake, has potent oxygen radical scavenging properties, activates the PXR receptor and has been associated with neuroprotection and with a lower risk for type 2 diabetes (29168502). These findings should be confirmed in larger cohorts from different continents.”.
- L295: Redundant statements. Please simplify the sentences.
R: The redundant statements were deleted.
- L304: Please use terminology correctly: 1) "probiotic property" is used for "probiotics" or "potential probiotics"; 2) "microbiota" includes all the microorganisms in a specific environmental sample. Please use "microbes"/"microorganisms " here.
R: Done as suggested
- L327: When describing the findings of a reference, please use the past tense and check for similar issues in this review!
- Done
- L331: Please be careful with this statement. I would suggest deleting "from commensal to pathogenic lifestyle". Same comments for L334 and L335
R: We thank the reviewer for point this out. This has now been modified as follows: “During stays in the ICU, the microbial diversity can be lost, and resistance genes can be selected. Most ultra-low-membership communities had low virulence when grouped together, however they showed a harmful behaviour when isolated. In ICU patients, the gut microbiota could be formed by ultra-low-diversity communities of multidrug-resistant pathogens and the shift from low to high virulence might be caused by opioids released during critical illness. Phosphate-polyethylene glycol [Pi-PEG] behaved as an antivirulence agent by increasing local phosphate availability and prevented opioid-induced virulence. Thus, local gut phosphate can drive the behaviour of bacterial communities”

Reviewer 2 Report
Section 2. CKD-MBD
in my opinion, the authors should further underline with a brief comment how the hyperphosphatemia and the parallel increase of FGF23, two fundamental events in the context of CKD-MBD, are respectively involved in the onset of vascular calcification as well as left venticular hypertrophy (see about it: Cozzolino et al Toxins (Basel). 2019 Apr 9;11(4):213. doi: 10.3390/toxins11040213. + Grabner et al Contrib Nephrol. Basel, Karger, 2017, vol 190, pp 83–95 (DOI: 10.1159/000468952)
Author Response
Comments and Suggestions for Authors
Section 2. CKD-MBD
- in my opinion, the authors should further underline with a brief comment how the hyperphosphatemia and the parallel increase of FGF23, two fundamental events in the context of CKD-MBD, are respectively involved in the onset of vascular calcification as well as left venticular hypertrophy (see about it: Cozzolino et al Toxins (Basel). 2019 Apr 9;11(4):213. doi: 10.3390/toxins11040213. + Grabner et al Contrib Nephrol. Basel, Karger, 2017, vol 190, pp 83–95 (DOI: 10.1159/000468952)
R: We thank the reviewer for the suggestion that has new been incorporated as follows. “Indeed, hyperphosphatemia and the parallel increase of FGF23 are respectively involved in the onset of vascular calcification as well as left ventricular hypertrophy (doi: 10.3390/toxins11040213. DOI: 10.1159/000468952.”

Reviewer 3 Report
The paper presents a current and important topic of CKD pathophysiology taking into account the role of microbiota. However, there are some concerns, that need addressing:
- writing style of the manuscript needs improvement (mainly within the Abstract and Conclusions sections),
- figure 1 and figure 2 should have higher quality, because now they are unclear.
- I do not understand the concept of Table 2. Is it really a Table, what is its purpose? Such key research questions should appear at the Introduction section. It would be more helpful to be able to read the one sentence summaries of the cited research instead - formed as a Table.
- Could the authors indicate any novelty of their paper? Does it add more to the existing knowledge. Why it is necessary and how it is better than earlier publications?
- Could you elaborate on PAOs more in your manuscript?
Author Response
The paper presents a current and important topic of CKD pathophysiology taking into account the role of microbiota. However, there are some concerns, that need addressing:
- writing style of the manuscript needs improvement (mainly within the Abstract and Conclusions sections),
R: the abstract and conclusion have been modified, especially the latter has been extensively modified.
- figure 1 and figure 2 should have higher quality, because now they are unclear
R: this has now been addressed
- I do not understand the concept of Table 2. Is it really a Table, what is its purpose? Such key research questions should appear at the Introduction section. It would be more helpful to be able to read the one sentence summaries of the cited research instead - formed as a Table.
R: as suggested, a table summary has been added as the new table 2. Additionally, the prior table 2 is now labelled table 3 and has been better integrated into the conclusions. We would respectfully keep table 3 at the end of the manuscript, as the questions arose from the review, based on the evidence gathered and represented in the manuscript.
- Could the authors indicate any novelty of their paper? Does it add more to the existing knowledge. Why it is necessary and how it is better than earlier publications?
R: We believe this is the first review that provides a holistic approach to the topic of phosphate and the gut microbiota in the context of kidney disease. An updated (March 27, 2021) PubMed search of the terms phosphate, microbiota and kidney did not disclose any similar review. This is now clearly indicated in the text.
- Could you elaborate on PAOs more in your manuscript?
R: As suggested, the section on PAOs has been extended and a new figure has bene added to clarify the concept.

Round 2
Reviewer 1 Report
Two more comments:
L220 Bacteroides are involved in the formation of butyrate, but there are more major producers in the intestine. Please add more information. And when referring to microbial genera/species, please use italics.
L344 Please check the definition of "probiotic" carefully: "Live microorganisms that, when administered in adequate amounts, confer a health benefit on the host". Ref: https://www.nature.com/articles/s41575-020-00390-5
Author Response
We thank the reviewer for carefully reading our text and pinpointing mistakes and typos
- a) Italics is now used throughout the text for genera and species
- b) The record for butyrate production has been set straight, by removing the refence to Bacteroides in line 200 and adding the key producers according to the enzyme used in line 252
- c) The typo for probiotics has now been corrected to prebiotic
